# OFF-POLICY ACTOR-CRITIC WITH SHARED EXPERIENCE REPLAY

## ABSTRACT

We investigate the combination of actor-critic reinforcement learning algorithms with uniform large-scale experience replay and propose solutions for two challenges: (a) efficient actor-critic learning with experience replay (b) stability of off-policy learning where agents learn from other agents behaviour. We employ those insights to accelerate hyper-parameter sweeps in which all participating agents run concurrently and share their experience via a common replay module.

To this end we analyze the bias-variance tradeoffs in *V-trace*, a form of importance sampling for actor-critic methods. Based on our analysis, we then argue for mixing experience sampled from replay with on-policy experience, and propose a new trust region scheme that scales effectively to data distributions where V-trace becomes unstable.

We provide extensive empirical validation of the proposed solution. We further show the benefits of this setup by demonstrating state-of-the-art data efficiency on Atari among agents trained up until 200M environment frames.

## 1 INTRODUCTION

Value-based and actor-critic policy gradient methods are the two leading techniques of constructing general and scalable reinforcement learning agents (Sutton et al., 2018). Both have been combined with non-linear function approximation (Tesauro, 1995; Williams, 1992), and have achieved remarkable successes on multiple challenging domains; yet, these algorithms still require large amounts of data to determine good policies for any new environment. To improve data efficiency, experience replay agents store experience in a memory buffer (replay) (Lin, 1992), and reuse it multiple times to perform reinforcement learning updates (Riedmiller, 2005). Experience replay allows to generalize prioritized sweeping (Moore & Atkeson, 1993) to the non-tabular setting (Schaul et al., 2015), and can also be used to simplify exploration by including expert (e.g., human) trajectories (Hester et al., 2017). Overall, experience replay can be very effective at reducing the number of interactions with the environment otherwise required by deep reinforcement learning algorithms (Schaul et al., 2015). Replay is often combined with the value-based Q-learning (Mnih et al., 2015), as it is an off-policy algorithm by construction, and can perform well even if the sampling distribution from replay is not aligned with the latest agent's policy. Combining experience replay with actor-critic algorithms can be harder due to their on-policy nature. Hence, most established actor-critic algorithms with replay such as (Wang et al., 2017; Gruslys et al., 2018; Haarnoja et al., 2018) employ and maintain Q-functions to learn from the replayed off-policy experience.

In this paper, we demonstrate that off-policy actor-critic learning with experience replay can be achieved without surrogate Q-function approximators using V-trace by employing the following approaches: a) off-policy replay experience needs to be mixed with a proportion of on-policy experience. We show experimentally (Figure 2) and theoretically that the V-trace policy gradient is otherwise not guaranteed to converge to a locally optimal solution. b) a trust region scheme (Conn et al., 2000; Schulman et al., 2015; 2017) can mitigate bias and enable efficient learning in a strongly off-policy regime, where distinct agents share experience through a commonly shared replay module. Sharing experience permits the agents to benefit from parallel exploration (Kretchmar, 2002) (Figures 1 and 3).

Our paper is structured as follows: In Section 2 we revisit pure importance sampling for actor-critic agents (Degris et al., 2012) and V-trace, which is notable for allowing to trade off bias and variance

in its estimates. We recall that variance reduction is necessary (Figure 4 left) but is biased in V-trace. We derive proposition 2 stating that off-policy V-trace is not guaranteed to converge to a locally optimal solution – not even in an idealized scenario when provided with the optimal value function. Through theoretical analysis (Section 3) and experimental validation (Figure 2) we determine that mixing on-policy experience into experience replay alleviates the problem. Furthermore we propose a trust region scheme (Conn et al., 2000; Schulman et al., 2015; 2017) in Section 4 that enables efficient learning even in a strongly off-policy regime, where distinct agents share the experience replay module and learn from each others experience. We define the trust region in policy space and prove that the resulting estimator is correct (i.e. estimates an improved return).

As a result, we present state-of-the-art data efficiency in Section 5 in terms of median human normalized performance across 57 Atari games (Bellemare et al., 2013), as well as improved learning efficiency on DMLab30 (Beattie et al., 2016) (Table 1).

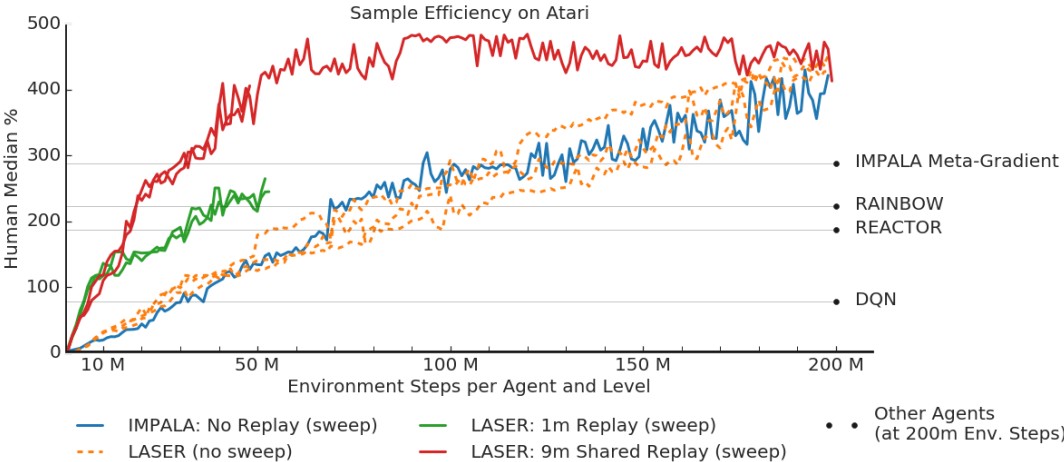

Figure 1: Sharing experience between agents leads to more efficient hyper-parameter sweeps on 57 Atari games. Prior art results are presented as horizontal lines (with scores cited from Gruslys et al. (2018), Hessel et al. (2017) and Mnih et al. (2013)). Note that the only previous agent "R2D2" that achieved a score beyond 400% required more than 3,000 million environment steps (see Kapturowski et al. (2019), page 14, Figure 9). We present the pointwise best agent from hyper-parameter sweeps with and without experience replay (shared and not shared). Each sweep contains 9 agents with different learning rate and entropy cost combinations. Replay experiment were repeated twice and ran for 50M steps. To report scores at 200M we ran the baseline and one shared experience replay agent for 200M steps.

Table 1: Comparison of state-of-the-art agents on 57 Atari games trained up until 200M environment steps (per game) and DMLab-30 trained until 10B steps (multi-task; all games combined). The first two rows are quoted from Xu et al. (2018) and Hessel et al. (2019), the third is our implementation of a pixel control agent from Hessel et al. (2019) and the last two rows are our proposed LASER (LArge Scale Experience Replay) agent. All agents use hyper-parameter sweeps expect for the marked.

| | Atari Median | DMLab-30 Median | DMLab-30 Mean-Capped |
|---|---|---|---|
| IMPALA Meta-Gradient (no sweep) | 287.6% at 200M | - | - |
| PopArt-IMPALA | - | - | 73.5% |
| PopArt-IMPALA+PixelControl | - | 85.5% | 77.6% |
| LASER: Experience Replay (no sweep) | 431% at 200M | | |
| LASER: Experience Replay | (233% at 50M) | 95.4% | 79.6% |
| LASER: Shared Experience Replay | (370% at 50M), 448% at 200M | 97.2% | 81.7% |

## 2 THE ISSUE WITH IMPORTANCE SAMPLING: BIAS AND VARIANCE IN V-TRACE

V-trace importance sampling is a popular off-policy correction for actor-critic agents (Espeholt et al., 2018). In this section we revisit how V-trace controls the (potentially infinite) variance that arises from naive importance sampling. We note that this comes at the cost of a biased estimate (see Proposition 1) and creates a failure mode (see Proposition 2) which makes the policy gradient biased. We discuss our solutions for said issues in Section 4.

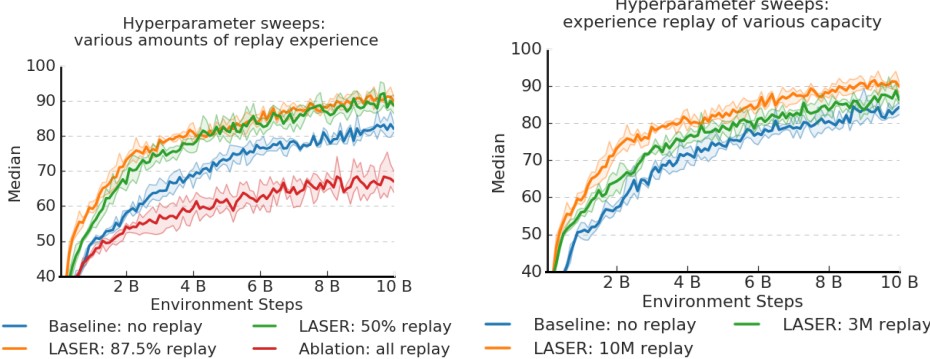

Figure 2: **Left**: Learning entirely off-policy from experience replay fails, while combining on-policy data with experience replay leads to improved data efficiency: We present sweeps on DMLab-30 with experience replays of 10M capacity. A ratio of $87.5\%$ implies that there are 7 replayed transitions in the batch for each online transition. Furthermore we consider an agent identical to "LASER $87.5\%$ replay" which however draws all samples from replay. Its batch thus does not contain any online data and we observe a significant performance decrease (see Proposition 2 and 3). The shading represents the point-wise best and worst replica among 3 repetitions. The solid line is the mean. **Right**: The effect of capacity in experience replay with $87.5\%$ replay data per batch on sweeps on DMLab-30. Data-efficiency improves with larger capacity.

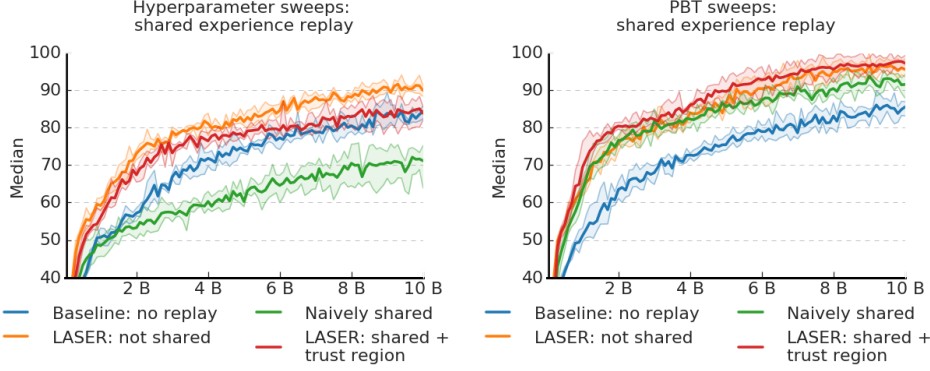

Figure 3: **Left:** Naively sharing experience between distinct agents in a hyper-parameter sweep fails (green) and is worse than the no-replay baseline (blue). The proposed trust region estimator mitigates the issue (red). **Right:** Combining population based training with trust region estimation improves performance further. All replay experiments use a capacity of 10 million observations and $87.5\%$ replay data per batch.

### 2.1 REINFORCEMENT LEARNING

We follow the notation of Sutton et al. (2018) where an *agent* interacts with its *environment*, to collect *rewards*. On each discrete time-step $t$, the agent selects an action $a_t$; it receives in return a reward $r_t$ and an *observation* $o_{t+1}$, encoding a partial view of the environment's *state* $s_{t+1}$. In the fully

observable case, the RL problem is formalized as a Markov Decision Process (Bellman, 1957): a tuple $(\mathcal{S}, \mathcal{A}, p, \gamma)$, where $\mathcal{S}, \mathcal{A}$ denotes finite sets of states and actions, $p$ models rewards and state transitions (so that $r_t, s_{t+1} \sim p(s_t, a_t)$), and $\gamma$ is a fixed discount factor. A *policy* is a mapping $\pi(a|s)$ from states to action probabilities. The agent seeks an *optimal* policy $\pi^*$ that maximizes the *value*, defined as the expectation of the cumulative discounted *returns* $G_t = \sum_{k=0}^{\infty} \gamma^k r_{t+k}$.

Off-policy learning is the problem of finding, or evaluating, a policy $\pi$ from data generated by a different policy $\mu$. This arises in several settings. Experience replay (Lin, 1992) mixes data from multiple iterations of policy improvement. In large-scale RL, decoupling acting from learning (Nair et al., 2015; Horgan et al., 2018; Espeholt et al., 2018) causes the experience to lag behind the latest agent policy. Finally, it is often useful to learn multiple general value functions (Sutton et al., 2011; Mankowitz et al., 2018; Lample & Chaplot, 2016; Mirowski et al., 2017; Jaderberg et al., 2017b) or *options* (Sutton et al., 1999; Bacon et al., 2017) from a single stream of experience.

## 2.2 NAIVE IMPORTANCE SAMPLING

On-policy n-step bootstraps give more accurate value estimates in expectation with larger $n$ (Sutton et al., 2018). They are used in many reinforcement learning agents (Mnih et al., 2016; Schulman et al., 2017; Hessel et al., 2017). Unfortunately $n$ must be chosen suitably as the estimates variance increases with $n$ too.

It is desirable to obtain benefits akin to n-step returns in the off-policy case. To this end multi-step importance sampling (Kahn, 1955) can be used. This however adds another source of (potentially infinite (Sutton et al., 2018)) variance to the estimate.

Importance sampling can estimate the expected return $V^\pi$ from trajectories sampled from $\mu \neq \pi$, as long as $\mu$ is non-zero whereever $\pi$ is. We employ a previously estimated value function $V$ as a bootstrap to estimate expected returns. Following Degris et al. (2012), a multi-step formulation of the expected return is

$$V^\pi(s_t) = \mathbf{E}_\mu \left[ V(s_t) + \sum_{k=0}^{K-1} \gamma^k \Big( \prod_{i=0}^{k} \frac{\pi_{t+i}}{\mu_{t+i}} \Big) \delta_{t+k} V \right] \tag{1}$$

where $\mathbf{E}_\mu$ denotes the expectation under policy $\mu$ up to an episode termination, $\delta_t V = r_t + \gamma V(s_{t+1}) - V(s_t)$ is the temporal difference error in consecutive states $s_{t+1}, s_t$, and $\pi_t = \pi_t(a_t|s_t)$. Importance sampling estimates can have high variance. Tree Backup (Precup et al., 2000), and Q($\lambda$) (Sutton et al., 2014) address this, but reduce the number of steps before bootstrapping even when this is undesirable (as in the on-policy case). RETRACE (Munos et al., 2016) makes use of full returns in the on-policy case, but it introduces a zero-mean random variable at each step, adding variance to empirical estimates in both on- and off-policy cases.

## 2.3 BIAS-VARIANCE ANALYSIS & FAILURE MODE OF V-TRACE IMPORTANCE SAMPLING

V-trace (Espeholt et al., 2018) reduces the variance of importance sampling by trading off variance for a biased estimate of the return – resulting in a failure mode (see Proposition 2). It uses clipped importance sampling ratios to approximate $V^\pi$ by $V^{\tilde\pi}(s_t) = V(s_t) + \sum_{k=0}^{K-1} \gamma^k \Big( \prod_{i=0}^{k-1} c_i \Big) \rho_t \delta_{t+k} V$ where $V$ is a learned state value estimate used to bootstrap, and $\rho_t = \min\left[ \pi_t/\mu_t, \bar\rho \right]$, $c_t = \min\left[ \pi_t/\mu_t, \bar c \right]$ are the clipped importance ratios. Note that, differently from RETRACE, V-trace fully recovers the Monte Carlo return when on policy. It similarly reweights the policy gradient as:

$$\nabla V^{\tilde\pi}(s_t) \stackrel{\text{def}}{=} \mathbf{E}_\mu \left[ \rho_t \nabla(\log \pi_t)(r_t + \gamma V^{\tilde\pi}(s_{t+1})) \right] \tag{2}$$

Note that $\nabla V^{\tilde\pi}(s_t)$ recovers the naively importance sampled policy gradient for $\bar\rho \to \infty$. In the literature, it is common to subtract a baseline from the action-value estimate $r_t + \gamma V^{\tilde\pi}(s_{t+1})$ to reduce variance (Williams, 1992), omitted here for simplicity. The constants $\bar\rho \geq \bar c \geq 1$ (typically chosen $\bar\rho = \bar c = 1$) define the level of clipping, and improve stability by ensuring a bounded variance. For any given $\bar\rho$, the bias introduced by V-trace in the value and policy gradient estimates increases with the difference between $\pi$ and $\mu$. We analyze this in the following propositions.

**Proposition 1.** *The V-trace value estimate $V^{\tilde\pi}$ is biased: It does not match the expected return of $\pi$ but the return of a related* implied *policy $\tilde\pi$ defined by equation 3 that depends on the behaviour*

*policy $\mu$:*

$$\tilde{\pi}_\mu(a|x) = \frac{\min\left[\bar{\rho}\mu(a|x), \pi(a|x)\right]}{\sum_{b \in A} \min\left[\bar{\rho}\mu(b|x), \pi(b|x)\right]} \tag{3}$$

*Proof.* See Espeholt et al. (2018). □

Note that the biased policy $\tilde{\pi}_\mu$ can be very different from $\pi$. Hence the V-trace value estimate $V^{\tilde{\pi}}$ may be very different from $V^\pi$ as well. As an illustrative example, consider two policies over a set of two actions, e.g. "left" and "right" represented as a tuple of probabilities. Let us investigate $\mu = (\phi, 1 - \phi)$ and $\pi = (1 - \phi, \phi)$ defined for any suitably small $\phi \leq 1$. Observe that $\pi$ and $\mu$ share no trajectories (state-action sequences) in the limit as $\phi \to 0$ and they get more focused on one action. A practical example of this could be two policies, one almost always taking a left turn and one always taking the right. Given sufficient data of either policy it is possible to estimate the value of the other e.g. with naive importance sampling. However observe that V-trace with $\bar{\rho} = 1$ will always estimate a biased value - even given infinite data. Observe that $\min\left[\mu(a|x), \pi(a|x)\right] = \min\left[\phi, 1 - \phi\right]$ for both actions. Thus $\tilde{\pi}_\mu$ is uniform rather than resembling $\pi$ the policy. The V-trace estimate $V^{\tilde{\pi}}$ would thus compute the average value of "left" and "right" – poorly representing the true $V^\pi$.

**Proposition 2.** *The V-trace policy gradient is biased: given the the optimal value function $V^*$ the V-trace policy gradient does not converge to a locally optimal $\pi^*$ for all off-policy behaviour distributions $\mu$.*

*Proof.* See Appendix C.

## 3 MIXING ON- AND OFF-POLICY EXPERIENCE

In Proposition 2 we presented a failure mode in V-trace where the variance reduction biases the value estimate and policy gradient. V-trace computes biased Q-estimates $Q^\omega \neq Q$ resulting in a wrong local policy gradient: $\nabla \mathbf{E}_{\pi(a|s)}\left[Q^\omega(s, a)\right] \neq \nabla \mathbf{E}_{\pi(a|s)}\left[Q(s, a)\right]$. In equation 10 we show that $Q^\omega(s, a) = Q(s, a)\omega(s, a)$ where $\omega(s, a) = \min\left[1, \bar{\rho}\frac{\mu(a|s)}{\pi(a|s)}\right] \leq 1$.

The question of how biased the resulting policy will be depends on whether the distortion changes the argmax of the Q-function. Little distortions that do not change the argmax will result in the same local fixpoint of the policy improvement. The policy will continue to select the optimal action and it will not be biased at this state. The policy will however be biased if the Q-function is distorted too much. For example consider a $\omega(s, a)$ that swaps the argmax for the 2nd largest value, the regret will then be the difference between the maximum and the 2nd largest value. Intuitively speaking the more distorted the $Q^\omega$, the larger will be the regret compared to the optimal policy.

More precisely, the regret of learning a policy that maximizes the distorted $Q^\omega$ at state $s$ is:

$$R(s) = Q(s, a^*) - Q(s, a_{\text{actual}}) = \max_b Q(s, b) - Q(s, a_{\text{actual}})$$

where $a^* = \text{argmax}_b(Q, b)$ is the optimal action according to the real $Q$ and $a_{\text{actual}} = \text{argmax}[Q^\omega(s, a)] = \text{argmax}[Q(s, a)\omega(s, a)]$, is the optimal action according to the distorted $Q^\omega$. For generality, we denote $A^*$ as the set of best actions - covering the case with multiple with identical optimal Q-values.

Proposition 3 provides a mitigation: Clearly the V-trace policy gradient will converge to the same solution as the true on-policy gradient if the argmax of the Q-function is preserved at all states in a tabular setting. We show that this can be achieved by mixing a sufficient proportion $\alpha$ of on-policy experience into the computation.

We show in equation 13 in the Appendix that choosing $\alpha$ such that

$$\frac{\alpha}{1-\alpha} > \max_{b \notin A^*}\left[\frac{Q^\omega(s, b) - Q^\omega(s, a^*)}{Q(s, a^*) - Q(s, b)}\right]\frac{d^\mu(s)}{d^\pi(s)} \text{ for } Q^\omega(s, a) = Q(s, a)\omega(s, a)$$

will result in a policy that correctly chooses the best action at state $s$. Note that $\frac{\alpha}{1-\alpha} \to \infty$ as $\alpha \to 1$.

Intuitively: the larger the action value gap of the real Q-function $Q(s, a^*) - Q(s, b)$ the lower the right hand side and the less on-policy data is required. If $\max_b[(Q(s, b)\omega(s, b) - Q(s, a^*)\omega(s, a^*)]$ is negative, then $\alpha$ may be as small as zero and we enabling even pure off-policy learning. Finally note that the right hand side decreases due to $d^\mu(s)/d^\pi(s)$ if $\pi$ visits the state $s$ more often than $\mu$.

All of those conditions can be computed and checked if an accurate Q-function and state distribution is accessible. How to use imperfect Q-function estimates to adaptively choose such an $\alpha$ remain a question for future research.

We provide experimental evidence for these results with function approximators in the 3-dimensional simulated environment DMLab-30 with various $\alpha \geq 1/8$ in Section 5.3 and Figure 2. We observe that $\alpha = 1/8$ is sufficient to facilitate stable learning. Furthermore it results in better data-efficiency than pure on-policy learning as it utilizes off-policy replay experience.

**Proposition 3.** *Mixing on-policy data into the V-trace policy gradient with the ratio $\alpha$ reduces the bias by providing a regularization to the implied state-action values. In the general function approximation case it changes the off-policy V-trace policy gradient from $\sum_s d^\mu(s)\mathbf{E}_\pi[(Q(s, a)\nabla \log \pi(a|s)]$ to $\sum_s \mathbf{E}_\pi[Q^\alpha(s, a)\nabla \log \pi(a|s)]$ where $Q^\alpha = Qd^\pi(s)\alpha + Q^\omega d^\mu(s)(1 - \alpha)$ is a regularized state-action estimate and $d^\pi$, $d^\mu$ are the state distributions for $\pi$ and $\mu$. Note that there exists $\alpha \leq 1$ such that $Q^\alpha$ has the same argmax (i.e. best action) as $Q$.*

*Proof.* See Appendix C.

Mixing online data with replay data has also been argued for by Zhang & Sutton (2017), as a heuristic way of reducing the sensitivity of reinforcement learning algorithms to the size of the replay memory. Proposition 3 grounds this in the theoretical properties of V-trace.

# 4 TRUST REGION SCHEME FOR OFF-POLICY V-TRACE

To mitigate the bias and variance problem of V-trace and importance sampling we propose a trust region scheme that adaptively selects only suitable behaviour distributions when estimating the state-value of $\pi$. To this end we introduce a *behaviour relevance function* that classifies behaviour as relevant. We then define a trust-region estimator that computes expectations (such as expected returns, or the policy gradient) only on relevant transitions. In proposition 4 and 5 we show that this trust region estimator indeed computes new state-value estimates that improve over the current value function. While our analysis and proof is general we propose a suitable behaviour relevance function in section 4.3 that employs the Kullback Leibler divergence between target policy $\pi$ and implied policy $\tilde{\pi}_\mu$: $\mathrm{KL}(\pi(\cdot|s)||\tilde{\pi}_\mu(\cdot|s))$. We provide experimental validation in Figure 3.

## 4.1 BEHAVIOUR RELEVANCE FUNCTIONS

In off-policy learning we often consider a family of behaviour policies either indexed by training iteration $t$: $M_T = \{\mu_t | t < T\}$ for experience replay, or by a different agent $k$: $M_K = \{\mu_k | k \in K\}$ when training multiple agents. In the classic experience replay case we then sample a time $t$ and locate the transition $\tau$ that was generated earlier via $\mu_t$. This extends naturally to the multiple agent case where we sample an agent index $k$ and then obtain a transition for such agent or tuples of $(k, t)$. Without loss of generality we simplify this notation and index sampled behaviour policies by a random variable $z \sim Z$ that represents the selection process. While online reinforcement learning algorithms process transitions $\tau \sim \pi$, off-policy algorithms process $\tau \sim \mu_z$ for $z \sim Z$. In this notation, given equation (1) and a bootstrap $V$, the expectation of importance sampled off-policy returns at state $s_t$ is described by:

$$V_{\mathrm{mix}}^\pi(s_t) = \mathbf{E}_z\Big[\mathbf{E}_{\mu_z|z}\big[G^{\pi,\mu_z}(s_t)\big]\Big] \tag{4}$$

where $G^{\pi,\mu}(s_t) = V(s_t) + \sum_{k=0}^\infty \gamma^k \Big(\prod_{i=0}^k \frac{\pi_{t+i}}{\mu_{t+i}}\Big)\delta_{t+k}V$ is a single importance sampled return. Note that the on-policy return $G^{\pi,\pi}(s_t) = V(s_t) + \sum_{k=0}^\infty \gamma^k r_{t+k}$.

Above $\mathbf{E}_{\mu_z|z}$ represents the expectation of sampling from a given $\mu_z$. The conditioning on $z$ is a notational reminder that this expectation does not sample $z$ or $\mu_z$ but experience from $\mu_z$. For any

sampled $z$ we obtain a $\mu_z$ and observe that the inner expectation wrt. experience of $\mu_z$ in equation (4) recovers the expected on-policy return in expectation:

$$
\begin{aligned}
\mathbf{E}_{\mu_z|z}\left[G^{\pi,\mu_z}(s_t)\right] &= \mathbf{E}_{\mu_z|z}\left[V(s_t) + \sum_{k=0}^{\infty} \gamma^k \Big(\prod_{i=0}^{k} \frac{\pi_{t+i}}{\mu_{z,t+i}}\Big)\delta_{t+k}V\right] \\
&= \mathbf{E}_{\pi}\left[V(s_t) + \sum_{k=0}^{\infty} \gamma^k \Big(\prod_{i=0}^{k} \frac{\mu_{z,t+i}}{\mu_{z,t+i}}\Big)\delta_{t+k}V\right] \quad (5) \\
&= \mathbf{E}_{\pi}\left[V(s_t) + \sum_{k=0}^{\infty} \gamma^k r_{t+k}\right] = \mathbf{E}_{\pi}\left[G^{\pi,\pi}(s_t)\right] = V^{\pi}(s_t)
\end{aligned}
$$

Thus $V^{\pi}_{\mathrm{mix}}(s_t) = \mathbf{E}_z\left[\mathbf{E}_{\pi}\left[G^{\pi,\pi}(s_t)\right]\right] = \mathbf{E}_{\pi}\left[G^{\pi,\pi}(s_t)\right] = V^{\pi}(s_t)$. This holds provided that $\mu_z$ is non-zero wherever $\pi$ is. This fairly standard assumption leads us straight to the core of the problem: it may be that some behaviours $\mu_z$ are ill-suited for estimating the inner expectation. However, standard importance sampling applied to very off-policy experience divides by small $\mu$ resulting in high or even infinite variance. Similarly, V-trace attempts to compute an estimate of the return following $\pi$ resulting in limited variance at the cost of a biased estimate in turn.

The key idea of our proposed solution is to compute the return estimate for $\pi$ at each state only from a subset of suitable behaviours $\mu_z$:

$$
M_{\beta,\pi}(s) = \{\mu_z | z \in Z \text{ and } \beta(\pi,\mu,s) < b\}
$$

as determined by a *behaviour relevance function* $\beta(\pi,\mu,s) : (M_Z, M_Z, S) \to \mathbb{R}$ and a threshold $b$. The behaviour relevance function decides if experience from a behaviour is suitable to compute an expected return for $\pi$. It can be chosen to control properties of $V^{\pi}_{\mathrm{mix}}$ by restricting the expectation on subsets of $Z$. In particular it can be used to control the variance of an importance sampled estimator: Observe that the inner expectation $\mathbf{E}_{\mu_z}\left[G^{\pi,\mu}(s_t)\big|z\right]$ in equation (4) already matches the expected return $V^{\pi}$. Thus we can condition the expectation on arbitrary subsets of $Z$ without changing the expected value of $V^{\pi}_{\mathrm{mix}}$. This allows us to reject high variance $G^{\pi,\mu}$ without introducing a bias in $V^{\pi}_{\mathrm{mix}}$. The same technique can be applied to V-trace where we can reject return estimates with high bias.

## 4.2 DERIVATION OF TRUST REGION ESTIMATORS

Using a behaviour relevance function $\beta(s)$ we can define a trust region estimator for regular importance sampling (IS) and V-trace and show their correctness.

We define the trust region estimator as the conditional expectation

$$
V^{\pi}_{\mathrm{trusted}}(s_t) = \mathbf{E}_z\left[\mathbf{E}_{\mu_z|z}\left[G^{\pi,\mu_z,\beta}(s_t)\right]\Big|\mu_z \in M_{\beta,\pi}(s_t)\right] \quad (6)
$$

with $\lambda$-returns $G$, chosen as $G_{\mathrm{IS}}$ for importance sampling and $G_{\mathrm{Vtrace}}$ for V-trace:

$$
G^{\pi,\mu_z}_{\mathrm{IS}}(s_t) = V(s_t) + \sum_{k=0}^{\infty} \gamma^k \Big(\prod_{i=0}^{k} \lambda_{\pi,\mu_z}(s_{t+i})\frac{\pi_{t+i}}{\mu_{z,t+i}}\Big)\delta_{t+k}V \quad (7)
$$

$$
G^{\pi,\mu_z}_{\mathrm{Vtrace}}(s_t) = V(s_t) + \sum_{k=0}^{\infty} \gamma^k \Big(\prod_{i=0}^{k-1} \lambda_{\pi,\mu_z}(s_{t+i})c_{z,t+i}\Big)\lambda_{\pi,\mu_z}(s_{t+k})\rho_{z,t+k}\delta_{t+k}V \quad (8)
$$

where $\lambda_{\pi,\mu}(s_t)$ is designed to constraint Monte-Carlo bootstraps to relevant behaviour: $\lambda_{\pi,\mu}(s_t) = \mathbb{1}_{\beta(\pi,\mu,s_t)<b}$ and $\rho_{z,t+k} = \min\left[\frac{\pi_{t+i}}{\mu_{z,t+i}}, \bar{\rho}\right]$ and $c_{z,t+k}$ are behaviour dependent clipped importance rations. Thus both $G^{\pi,\mu_z}_{\mathrm{IS}}$ and $G^{\pi,\mu_z}_{\mathrm{Vtrace}}$ are a multi-step return estimators with adaptive length. Note that only estimators with length $\geq 1$ are used in $V^{\pi}_{\mathrm{trusted}}$. Due to Minkowski's inequality the trust region estimator thus shows at least the same contraction as a 1-step bootstrap, but can be faster due to its adaptive nature:

**Proposition 4.** *Let $G^{\pi,\mu_z}_{\mathrm{IS}}$ be a set of importance sampling estimators as defined in equation 7. Note that they all have the same fix point $V^{\pi}$ and contract with at least $\gamma$. Then the contraction properties carry over to $V^{\pi}_{\mathrm{trusted}}$. In particular $|V^{\pi}_{\mathrm{trusted}} - V^{\pi}|_{\infty} \leq \gamma |V - V^{\pi}|_{\infty}$.*

*Proof.* See Appendix C.

**Proposition 5.** *Let $G_{\text{Vtrace}}^{\pi,\mu_z}$ be a set of V-trace estimators (see equation 8) with corresponding fixed points $V^z$ (see equation 3) to which they contract at a speed of an algorithm and behaviour specific $\eta_z$. Then $V_{\text{trusted}}^{\pi}$ moves towards $V^{\beta} = \mathbf{E}_{z|\mu_z \in M_{\beta,\pi}(s_t)}[V^z]$ shrinking the distance as follows $\left|V_{\text{trusted}}^{\pi} - V^{\beta}\right|_{\infty} < \max_{\mu_z \in M_{\beta,\pi}(s_t)} |\eta_z(V - V^z)|_{\infty} \leq \eta_{\max} \max_{\mu_z \in M_{\beta,\pi}(s_t)} |(V - V^z)|_{\infty}$ with $\eta_{\max} = \max_{\mu_z \in M_{\beta,\pi}(s_t)} \eta_z$.*

*Proof.* See Appendix C.

Note how the choice of $\beta$ and thus $M_{\beta,\pi}$ enables us to discard ill-suited $G_{\text{Vtrace}}^{\pi,\mu_z}$ from the estimation of $V_{\text{trusted}}^{\pi}$. Recall that V-trace fixed points $V_z$ are biased. Thus $\beta$ allows us to selectively create the V-trace target $V^{\beta} = \mathbf{E}_{z|\mu_z \in M_{\beta,\pi}(s_t)}[V^z]$ and control its bias and the shrinkage $\max_{\mu_z \in M_{\beta,\pi}(s_t)} |\eta_z(V(s) - V^z(s))|_{\infty}$ (see Proposition 5). Similarly it can control cases where we can not use the exact importance sampled estimator. The same approach based on nested expectations can be applied to the expectation of the policy gradient estimate and allows to control the bias and greediness (see Proposition 2) there as well.

### 4.3 IMPLEMENTATION DETAILS

In Proposition 5 we have seen that the quality of the trust region V-trace return estimator depends on $\beta$. A suitable choice of $\beta$ can move the return estimate $V^{\beta}$ closer to $V^{\pi}$ and improve the shrinkage by reducing $\max_{\mu_z \in M_{\beta,\pi}(s_t)} |\eta_z(V(s) - V^z(s))|_{\infty}$. Hence, we employ a behaviour relevance function $\beta_{\text{KL}}$ that rejects high bias transitions by estimating the Kulback-Leibler divergence between the target policy $\pi$ and the implied policy $\tilde{\pi}_{\mu_z}$ for a sampled behaviour $\mu_z$. Recall from Proposition 1 that $\tilde{\pi}_{\mu_z}$ determines the fixed point of the V-trace estimator for behaviour $\mu_z$ and thus determines the bias in $V^z$.

$$\beta_{\text{KL}}(\pi, \mu, s) = \text{KL}\left(\pi(\cdot|s)||\tilde{\pi}_{\mu}(\cdot|s)\right)$$

Note that the behaviour probabilities $\mu_z$ can be evaluated and saved to the replay when the agent executes the behaviour, similarly the target policy $\pi$ is represented by the agents neural network. Using both and equation 3, $\tilde{\pi}_{\mu}$ can be computed. For large or infinite action spaces a Monte Carlo estimate of the KL divergence can be computed.

It is possible to define separate behaviour relevance functions for the policy and value estimate. For simplicity we reject transitions entirely for all estimates and do not consider rejected transitions for the policy gradient and value gradient updates or auxiliary tasks. As described above we stop the Monte-Carlo bootstraps once they reach undesirable state-behaviour pairs. Note that this censoring procedure is computed from state dependent $\beta(\pi, \mu, s)$ and ensures that the choice of bootstrapping does not depend on the sampled actions. Note that rejection by an action-based criteria such as small $\pi(a|s)/\mu(a|s)$ would introduce an additional bias which we avoid by choosing $\beta_{\text{KL}}$.

## 5 EXPERIMENTS

We present experiments to support the following claims:

- Section 5.2: Uniform experience replay obtains comparable results as prioritized experience replay, while being simpler to implement and tune.

- Section 5.3: Using fresh experience before inserting it in experience replay is better than learning purely off-policy from experience replay – in line with Proposition 3.

- Section 5.4: Sharing experience without trust region performs poorly as suggested by Proposition 2. Off-Policy Trust-Region V-trace solves this issue.

- Section 5.5: Sharing experience can take advantage of parallel exploration and obtains state-of-the-art performance on Atari games, while also saving memory through sharing a single experience replay.

## 5.1 Experimental Setup & Methodology

We use the V-trace distributed reinforcement learning agent (Espeholt et al., 2018) as our baseline. In our experiments we consider two experimental platforms: Atari and DeepMind Lab. On Atari we consider the common single task training regime, where a different agent is trained, from scratch, on each of the tasks. Following Xu et al. (2018) we use a discount of 0.995. Motivated by recent work by Kaiser et al. (2019), we use the IMPALA deep network and increased the number of channels $4\times$. We use $96\%$ replay data per batch. Differently from Espeholt et al. (2018), we do not use gradient clipping by norm (Pascanu et al., 2012). Updates are computed on mini-batches of 32 (regular) and 128 (replay) trajectories, each corresponding to 19 steps in the environment. In the context of DeepMind Lab, we consider the multi-task suite DMLab-30 (Espeholt et al., 2018), as the visuals and the dynamics are more consistent across tasks. Furthermore the multi-task regime is particularly suitable for the investigation of strongly off-policy data distributions arising from sharing the replay across agents, as concurrently learning agents can easily be stuck in different policy plateaus, generating substantially different data (Schaul et al., 2019). As in Espeholt et al. (2018), in the multi-task setting each agent trains simultaneously on a uniform mixture of all tasks rather than individually on each game. The score of an agent is thus the median across all 30 tasks. Following Hessel et al. (2019), we augment our agent with multi-task Pop-Art normalization and PixelControl. We use a PreCo LSTM (Amos et al., 2018) instead of the vanilla one (Hochreiter & Schmidhuber, 1997). Updates are computed on mini-batches of multiple trajectories chosen as above, each corresponding to 79 steps in the environment. In early experiments we found that computing the entropy cost only on the online data provided slightly better results, hence we have done so throughout our experiments.

In all our experiments, experience sampled from memory is mixed with online data within each mini-batch – following Proposition 3. Episodes are removed in a first in first out order, so that replay always holds the most recent experience. Unless explicitly stated otherwise we consider hyper-parameter sweeps, some of which share experience via replay. In this setting multiple agents start from-scratch, run concurrently at identical speed, and add their new experience into a common replay buffer. All agents will then draw uniform samples from the replay buffer. On DMLab-30 we consider both regular hyper-parameter sweeps and sweeps with population based training (PBT) (Jaderberg et al., 2017a). On DMLab-30 sweeps contain 10 agents with hyper-parameters sampled similar as Espeholt et al. (2018) but fixed RMSProp $\epsilon = 0.1$. On Atari sweeps contain 9 agents with different constant learning rate and entropy cost combinations $\{3 \cdot 10^{-4}, 6 \cdot 10^{-4}, 1.2 \cdot 10^{-3}\} \times \{5 \cdot 10^{-3}, 1 \cdot 10^{-2}, 2 \cdot 10^{-2}\}$ (distributed by factors $\{1/2, 1, 2\}$ around the initial parameters reported in Espeholt et al. (2018)). Although our focus is on efficient hyper-parameter sweeps given crude initial parameters, we also present a single-agent LASER experiment using the same tuned schedule as Espeholt et al. (2018), a $87.5\%$ replay ratio and a 15M replay. We store the entire episodes in the replay buffer and replay each episode from the beginning, using the most recent network parameters to recompute the LSTM states along the way: this is particularly critical when sharing experience between different agents, which may have arbitrarily different state representations.

## 5.2 Uniform and Prioritized Experience Replay

Prioritized experience replay has the potential to provide more efficient learning compared to uniform experience replay (Schaul et al., 2015; Horgan et al., 2018). However, it also introduces a number of new hyper-parameters and design choices: the most critical are the priority metric, how strongly to bias the sampling distribution, and how to correct for the resulting bias. Uniform replay is instead almost parameter-free, requires little tuning and can be easily shared between multiple agents. Experiments provided in Figure 4 in the appendix showed little benefit of actor critic prioritized replay on DMLab-30. Furthermore priorities are typically computed from the agent specific metrics such as the TD-error, which are ill-defined when replay is shared among multiple agents. Hence we used uniform replay for our further investigations.

## 5.3 Mixing On- and Off-policy Experience and Replay Capacity

Figure 2 (left) shows that performance degrades significantly when online data is not present in the batch. This experimentally validates Propositions 2 and 3 that highlight difficulties of learning purely off-policy. Furthermore Figure 2 (right) shows that best results are obtained with experience replay

of 10M capacity and $87.5\%$ ratio. A ratio of $87.5\% = 7/8$ corresponds to 7 replay samples for each online sample. We have considered ratios of $1/2, 3/4$, and $7/8$ and observed stable training for all of them. Observe that among those values, larger ratios are more data-efficient as they take advantage of more replayed experience per training step.

## 5.4 SHARED EXPERIENCE REPLAY WITH OFF-POLICY TRUST REGION V-TRACE

In line with proposition 2 we observe in Figure 3 (left) that hyper-parameter sweeps without trust-region are even surpassed by the baseline without experience replay. State-of-the-art results are obtained in Figure 3 (right) when experience is shared with trust-region in a PBT sweep.

Observe that this indicates parallel exploration benefits and saves memory at the same time: in our sweep of 10 replay agents the difference between $10 \times 10M$ (separate replays) and 10M (shared replay) is 10-fold. This effect would be even more pronounced with larger sweeps.

As discussed in section 2.3, the bias in V-trace occurs due to the clipping of importance ratios. A potential solution of reducing the bias would be to increase the $\bar{\rho}$ threshold to clip less aggressively and accept increased variance. Figure 4 in the appendix shows that this is not a solution.

## 5.5 EVALUATION ON ATARI

We apply our proposed agent to Atari which has been a long established suite to evaluate reinforcement algorithms (Bellemare et al., 2013). Since we focus on sample-efficient learning we present our results in comparison to prior work at 200M steps (Figure 1). Shared experience replay obtains even better performance than not shared experience replay. This confirms the efficient use of parallel exploration (Kretchmar, 2002). The fastest prior agent to reach $400\%$ is presented by Kapturowski et al. (2019) requiring more than 3,000M steps. LASER with shared replay achieves $423\%$ in 60M per agent. Given 200M steps it achieves $448\%$. We also present a single (no sweep) LASER agent that achieves $431\%$ in 200M steps.

## 6 CONCLUSION

We have presented LASER – an off-policy actor-critic agent which employs a large and shared experience replay to achieve data-efficiency. By sharing experience between concurrently running experiments in a hyper-parameter sweep it is able to take advantage of parallel exploration. As a result it achieves state-of-the-art data efficiency on 57 Atari games given 200M environment steps. Furthermore it achieves competitive results on both DMLab-30 and Atari under regular, not shared experience replay conditions.

To facilitate this algorithm we have proposed two approaches: a) mixing replayed experience and on-policy data and b) a trust region scheme. We have shown theoretically and demonstrated through a series of experiments that they enable learning in strongly off-policy settings, which present a challenge for conventional importance sampling schemes.

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

## APPENDIX A    ADDITIONAL EXPERIMENTS

### A.1    REDUCED CLIPPING IN V-TRACE DOES NOT ENABLE SHARED EXPERIENCE REPLAY

Increasing the clipping constant $\bar{\rho}$ in V-trace reduces bias in favour of increased variance. We investigate if reducing bias in this manner enables sharing experience replay between multiple agents in a hyper-parameter sweep. Figure 4 (left) shows that this is not a solution, thus motivating our trust region scheme.

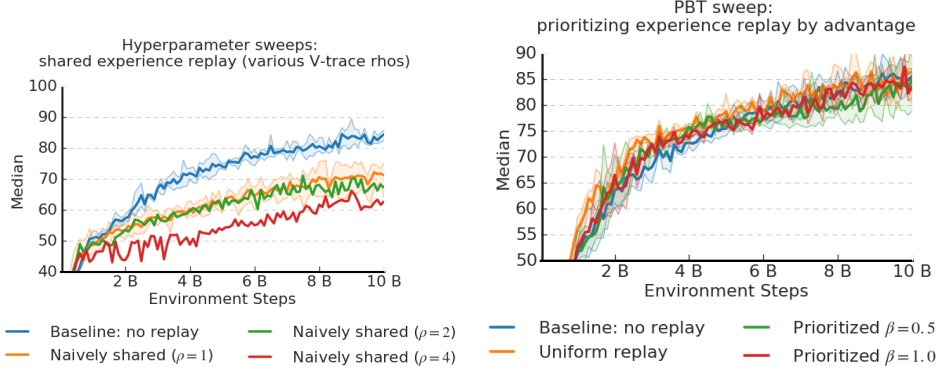

Figure 4: **Left:** Increasing the V-trace clipping constant $\bar{\rho}$ does not enable shared experience replay. In fact sharing experience replay in this particular way is worse than pure online learning. This motivates the use of our proposed trust region scheme. On a side note, increased clipping thresholds resulting in worse performance verifies the importance of variance reduction through clipping. **Right:** Median human normalized performance across 30 tasks for the best agent in a sweep, averaged across 2 replicas. All replay experiments use $50\%$ replay ratio and a capacity of 3 million observations. We investigate if uncorrected LSTM states can be used in combination with different replay modes. We consider uniform sampling and prioritization via the critic's loss, and include both full ($\beta = 1$) and partial ($\beta = 0.5$) importance corrections

## A.2 PRIORITIZED AND UNIFORM EXPERIENCE REPLAY, LSTM STATES

With prioritized experience replay each transition $\tau$ is sampled with probability $P(\tau) \propto p_\tau^\alpha$, for a suitable unnormalized priority score $p_\tau$ and a global tunable parameter $\alpha$. It is common (Schaul et al., 2015; Horgan et al., 2018; Hessel et al., 2017) to then weight updates computed from that sample by $1/P(\tau)^\beta$ for $0 < \beta \le 1$, where $\beta = 1$ fully corrects for the bias introduced in the state distribution. In one step temporal difference methods, typical priorities are based on the immediate TD-error, and are typically recomputed after a transition is sampled from replay. This means low priorities might stay low and get stale – even if the transition suddenly becomes relevant. To alleviate this issue, the sampling distribution is mixed with a uniform, as controlled by a third hyper parameter $\epsilon$.

The performance of agents with prioritized experience replay can be quite sensitive to the hyper-parameters $\alpha$, $\beta$, and $\epsilon$.

A critical practical consideration is how to implement random access for recurrent memory agents such as agents using an LSTM. Prioritized agents sample a presumably interesting transition from the past. This transition may be at any position within the episode. To infer the correct recurrent memory-state at this environment-state all earlier environment-states within that episode would need to be replayed. A prioritized agent with a random access pattern would thus require costly LSTM refreshes for each sampled transition. If LSTM states are not recomputed representational missmatch (Kapturowski et al., 2019) occurs.

Sharing experience between multiple agents amplifies the issue of LSTM state representation miss-match. Here each agent has its own network parameters and the state representations between agents may be arbitrarily different.

As a mitigation Kapturowski et al. (2019) use a burn-in window or to initialize with a constant starting state. We note that those solutions can only partially mitigate the fundamental issue and that counter examples such as arbitrarily long T-Mazes (Tolman, 1948; Olton, 1979) can be constructed easily.

We thus advocate for uniform sampling. In our implementation we uniformly sample an episode. Then we replay each episode from the beginning, using the most recent network parameters to recompute the LSTM states along the way: this is particularly critical when sharing experience between different agents, which may have arbitrarily different state representations.

This solution is exact and cost-efficient as it only requires one additional forward pass for each learning step (forward + backward pass).

An even more cost efficient approach would be to not refresh LSTM states at all. Naturally this comes at the cost of representational missmatch. However it would allow for an affordable implementation of prioritized experience replay. We investigate this in Figure 4 (right) and observe that it is not viable. We compare a baseline V-trace agent with no experience replay, one with uniform experience replay, and two different prioritized replay agents. We do not refresh LSTM states for any of the agents.

The uniform replay agent is more data efficient then the baseline, and also saturates at a higher level of performance. The best prioritized replay agent uses full importance sampling corrections ($\beta = 1$). However it performs no higher than with uniform replay. We therefore we used uniform replay with full state correction for all our investigations in the paper.

## A.3 EVALUATION PROTOCOL

For evaluation, we average episode returns within buckets of 1M (Atari) and 10M (DMLab) environment steps for each agent instance, and normalize scores on each game by using the scores of a human expert and a random agent (van Hasselt et al., 2016). In the multi-task setting, we then define the performance of each agent as the median normalized score of all levels that the agent trains on. Given the use of population based training, we need to perform the comparisons between algorithms at the level of sweeps. We do so by selecting the best performing agent instance within each sweep at any time. Note that for the multi-task setting, our approach of first averaging across many episodes, then taking the median across games, on DMLab further downsampling to 100M env steps, and only finally selecting the maximum within the sweep, results in substantially lower variance than if we were to compute the maximum before the median and smoothing.

All DMLab-30 sweeps are repeated $3\times$ with the exception of $\rho = 2$ and $\rho = 4$ in Figure 4. We then plot a shaded area between the point-wise best and worst replica and a solid line for the mean. Atari sweeps having 57 games are summarized and plotted by the median of the human-normalized scores.

## APPENDIX B    ALGORITHM PSEUDOCODE

We present algorithm pseudocode for LASER with trust region (Algorithm 1). For clarity we present a version without LSTM and focus on the single agent case. The multi-agent case is a simple extension where all agents save to the same replay database and also sample from the same replay. Also each agent starts with different network parameters and hyper-parameters. The LSTM state recomputation can be achieved with *Replayer Threads* (nearly identical to Actor Threads) that sample entire epsiodes from replay, step through them while reevaluating the LSTM state and slice the experience into trajectories of length $T$. Similar to regular LSTM Actor Threads from Espeholt et al. (2018) the Replayer Threads send each trajectory together with an LSTM state to the learning thread via a queue. The Learner Thread initializes the LSTM with the transmitted state when the LSTM is unrolled over the trajectory.

---

**Algorithm 1** Single Agent LASER with Trust Region

---

Initialize parameter vectors $\theta$. Initialize $\pi_1 = \pi_\theta$.
**Actor Thread:**
**while** training is ongoing **do**
    Sample trajectory unroll $u = \{\tau_t\}_{t \in \{1,\ldots,T\}}$ of length $T$ by acting in the environment using the latest $\pi_k$ where $\tau_t = (s_t, a_t, r_t, \mu_t = \pi_k(s_t|\cdot))$.
    Enqueue $u$ into *Lerner Queue*, wait if full.
    Add $u$ into *Replay Database*.
    Remove oldest trajectory if database has reached desired capacity limit.
**end while**
**Learner Thread:**
Given: Batch size $B$, online fraction $\alpha$.
**for** training iteration $k$ **do**
    Form training batch $U = \{u_b\}_{b \in \{1,\ldots,B\}}$ of $B$ trajectories of length $T$, by dequeuing $B\alpha$ trajectories from *Lerner Queue* and sampling $B(1 - \alpha)$ trajectories from *Replay Database*.
    Evaluate the target policy $\pi_k$ on the sampled transitions in $U$: i.e. $\pi_k(s_{b,t}|\cdot)$.
    Compute behaviour relevance mask $M$ with $M_{b,t} = \mathrm{KL}(\pi_k(s_{b,t}|\cdot)||\mu_{b,t}) < b$ where $\mu_{b,t}, s_{b,t}$ are obtained from $U_{b,t}$.
    Compute trust-region V-trace return $V_{t,b}$ using 8 where $\lambda_{\pi,\mu}(s_{b,t}) = M_{b,t}$.
    Let $[L_V(\theta)]_{t,b} = \frac{1}{2}(V_{t,b} - V_\theta(s_{t,b}))^2$.
    Let $A_{t,b} = V_{t,b} - V_\theta(s_{t,b})$ and $[L_P(\theta)]_{t,b} = \rho_{t,b} \log[\pi_\theta(s_{t,b}|a_{t,b})]A_{t,b}$, where $\rho$ is the clipped v-trace importance sampling ratio.
    Perform gradient update to $\theta$ using $\nabla_\theta \sum_{t,b}[L_V(\theta) + L_P(\theta)]_{t,b}M_{t,b}$, denote the resulting $\pi_\theta$ as $\pi_{k+1}$.
**end for**

---

## APPENDIX C    PROPOSITIONS

We have stated five propositions in our paper for which we provide proofs below.

**Proposition 1.** *The V-trace value estimate $V^{\tilde{\pi}}$ is biased: It does not match the expected return of $\pi$ but the return of a related* implied policy $\tilde{\pi}$ *defined by equation 9 that depends on the behaviour policy $\mu$:*

$$\tilde{\pi}_\mu(a|x) = \frac{\min[\bar{\rho}\mu(a|x), \pi(a|x)]}{\sum_{b \in A} \min[\bar{\rho}\mu(b|x), \pi(b|x)]} \tag{9}$$

*Proof.* See Espeholt et al. (2018). □

**Proposition 2.** *The V-trace policy gradient is biased: given the the optimal value function $V^*$ the V-trace policy gradient does not converge to a locally optimal $\pi^*$ for all off-policy behaviour distributions $\mu$.*

*Proof.* Proof by contradiction:

Consider a tabular counter example with a single (locally) optimal policy at $s_t$ given by $\pi^*(s_t) = \text{argmax}_\pi \left[ \sum_{a \in A} \pi(a|s_t)Q^*(a, s_t) \right]$ that always selects the action $\text{argmax}_a Q^*(a, s_t)$.

Even in this ideal tabular setting V-trace policy gradient estimates a different $\tilde{\pi}^*$ rather than the optimal $\pi^*$ as follows

$$
\begin{aligned}
\nabla V^{*,\pi}(s_t) &= \mathbf{E}_\mu \left[ \rho_t(r_t + \gamma V^*(s_{t+1})\nabla \log \pi(a_t|s_t)] \right. \\
&= \mathbf{E}_\mu \left[ \rho_t Q^*(s_t, a_t)\nabla \log \pi(a_t|s_t)] \right. \\
&= \mathbf{E}_\mu \left[ \min\left[ \frac{\pi(a_t|s_t)}{\mu(a_t|s_t)}, \bar{\rho} \right] Q^*(s_t, a_t)\nabla \log \pi(a_t|s_t) \right] \\
&= \mathbf{E}_\mu \left[ \frac{\pi(a_t|s_t)}{\mu(a_t|s_t)}\min\left[ 1, \bar{\rho}\frac{\mu(a_t|s_t)}{\pi(a_t|s_t)} \right] Q^*(s_t, a_t)\nabla \log \pi(a_t|s_t) \right] \quad (10) \\
&= \mathbf{E}_\pi \left[ \min\left[ 1, \bar{\rho}\frac{\mu(a_t|s_t)}{\pi(a_t|s_t)} \right] Q^*(s_t, a_t)\nabla \log \pi(a_t|s_t) \right] \\
&= \mathbf{E}_\pi \left[ \omega(s_t, a_t)Q^*(s_t, a_t)\nabla \log \pi(a_t|s_t) \right] \\
&= \mathbf{E}_\pi \left[ Q^{*,\omega}(s_t, a_t)\nabla \log \pi(a_t|s_t) \right]
\end{aligned}
$$

Observe how the optimal Q-function $Q^*$ is scaled by $\omega(s_t, a_t) = \min\left[ 1, \bar{\rho}\frac{\mu(a_t|s_t)}{\pi(a_t|s_t)} \right] \leq 1$ resulting in *implied state-action values* $Q^{*,\omega}$. This penalizes actions where $\mu(a_t|s_t)\bar{\rho} < \pi(a_t|s_t)$ and makes V-trace greedy w.r.t. to the remaining ones. Thus $\mu$ can be chosen adversarially to corrupt the optimal state action value. Note that $\bar{\rho}$ is a constant typically chosen to be 1.

To prove the lemma consider a counter example such as an MDP with two actions and $Q^* = (2, 5)$ and $\mu = (0.9, 0.1)$ and initial $\pi = (0.5, 0.5)$. Here the second action with expected return 5 is clearly favourable. Abusing notation $\mu/\pi = (1.8, 0.2)$. Thus $Q^{\tilde{\pi},\omega} = (2*1, 5*0.2) = (2, 1)$. Therefore $\tilde{\pi}^* = (1, 0)$ wrongly selects the first action. $\square$

**Proposition 3.** *Mixing on-policy data into the V-trace policy gradient with the ratio $\alpha$ reduces the bias by providing a regularization to the implied state-action values. In the general function approximation case it changes the off-policy V-trace policy gradient from $\sum_s d^\mu(s)\mathbf{E}_\pi \left[ (Q(s, a)\nabla \log \pi(a|s)] \right]$ to $\sum_s \mathbf{E}_\pi \left[ Q^\alpha(s, a)\nabla \log \pi(a|s)] \right]$ where $Q^\alpha = Qd^\pi(s)\alpha + Q^\omega d^\mu(s)(1 - \alpha)$ is a regularized state-action estimate and $d^\pi$, $d^\mu$ are the state distributions for $\pi$ and $\mu$. Note that there exists $\alpha \leq 1$ such that $Q^\alpha$ has the same argmax (i.e. best action) as $Q$.*

*Proof.* Note that the on-policy policy gradient is given by

$$
\nabla J_{\text{on}}(\pi) = \sum_s d^\pi(s)\mathbf{E}_\pi \left[ Q(s, a)\nabla \log \pi(a|s) \right]
$$

Similarly the off-policy V-trace gradient is given by

$$
\nabla J_{\text{off}}(\pi) = \sum_s d^\mu(s)\mathbf{E}_\pi \left[ \omega(s, a)Q(s, a)\nabla \log \pi(a|s) \right]
$$

with the V-trace distortion factor $\omega(s_t, a_t) = \min\left[ 1, \bar{\rho}\frac{\mu(a_t|s_t)}{\pi(a_t|s_t)} \right] \leq 1$ that can de-emphasize action values and $Q^\omega(s, a) = \omega(s, a)Q(s, a)$.

The $\alpha$-interpolation of both gradients can be transformed as follows:

$$
\begin{aligned}
\nabla\big(\alpha J_{\text{on}} + (1-\alpha) J_{\text{off}}\big)(\pi) &= \alpha \sum_s d^\pi(s) \mathbf{E}_\pi \left[ Q(s,a) \nabla \log \pi(a|s) \right] \\
&\quad + (1-\alpha) \sum_s d^\mu(s) \mathbf{E}_\pi \left[ \omega(s,a) Q(s,a) \nabla \log \pi(a|s) \right] \\
&= \sum_s d^\pi(s) \mathbf{E}_\pi \left[ Q(s,a) \alpha \nabla \log \pi(a|s) \right] \\
&\quad + \sum_s d^\mu(s) \mathbf{E}_\pi \left[ Q^\omega(s,a)(1-\alpha) \nabla \log \pi(a|s) \right] \\
&= \sum_s \mathbf{E}_\pi \left[ Q(s,a) d^\pi(s) \alpha \nabla \log \pi(a|s) \right] \\
&\quad + \sum_s \mathbf{E}_\pi \left[ Q^\omega(s,a) d^\mu(s)(1-\alpha) \nabla \log \pi(a|s) \right] \\
&= \sum_s \mathbf{E}_\pi \left[ Q(s,a) d^\pi(s) \alpha \nabla \log \pi(a|s) + Q^\omega(s,a) d^\mu(s)(1-\alpha) \nabla \log \pi(a|s) \right] \\
&= \sum_s \mathbf{E}_\pi \left[ \big( Q(s,a) d^\pi(s) \alpha + Q^\omega(s,a) d^\mu(s)(1-\alpha) \big) \nabla \log \pi(a|s) \right] \\
&= \sum_s \mathbf{E}_\pi \left[ Q^\alpha(s,a) \nabla \log \pi(a|s) \right]
\end{aligned}
\tag{11}
$$

for $Q^\alpha(s,a) = Q(s,a) d^\pi(s) \alpha + Q^\omega(s,a) d^\mu(s)(1-\alpha)$ $\qquad\qquad\square$

**Interpretation of Proposition 3**  As discussed in section 3 the V-trace policy gradient will have the correct local fixpoint at state $s$ if the argmax of the state-value function is preserved despite the distortion: i.e. if $\operatorname{argmax}_a[Q(s,a)] = \operatorname{argmax}_a[Q^\omega(s,a)]$. Respectively when mixing in an $\alpha \in [0,1)$ share of online data the fixpoint will be preserved if

$$
\operatorname{argmax}_a[Q(s,a)] = \operatorname{argmax}_a[Q^\alpha(s,a)]
\tag{12}
$$

Let $a^* = \operatorname{argmax}_b(Q,b)$ be any best action and $A^*$ be set of best actions. Then equation 12 is equivalent to:

$$
Q^\alpha(s,a^*) > Q^\alpha(s,b) \; \forall b \notin A^*
$$

Using the definition of $Q^\alpha$ this can be rewritten as:

$$
Q(s,a^*) d^\pi(s) \alpha + Q^\omega(s,a^*) d^\mu(s)(1-\alpha) > Q(s,b) d^\pi(s) \alpha + Q^\omega(s,b) d^\mu(s)(1-\alpha) \; \forall b \notin A^*
$$

Which can be rearranged to:

$$
[Q(s,a^*) d^\pi(s) - Q(s,b) d^\pi(s)] \alpha > [Q^\omega(s,b) d^\mu(s) - Q^\omega(s,a^*) d^\mu(s)](1-\alpha) \; \forall b \notin A^*
$$

By definition $Q(s,a^*) d^\pi(s) - Q(s,b) d^\pi(s) > 0 \; \forall b \notin A^*$, hence:

$$
\frac{\alpha}{1-\alpha} > \frac{Q^\omega(s,b) - Q^\omega(s,a^*)}{Q(s,a^*) - Q(s,b)} \frac{d^\mu(s)}{d^\pi(s)} \; \forall b \notin A^*
$$

It follows that the policy gradient will have the same local fixpoint if

$$
\frac{\alpha}{1-\alpha} > \max_{b \notin A^*} \left[ \frac{Q^\omega(s,b) - Q^\omega(s,a^*)}{Q(s,a^*) - Q(s,b)} \right] \frac{d^\mu(s)}{d^\pi(s)}
\tag{13}
$$

Note that $\frac{\alpha}{1-\alpha} \to \infty$ as $\alpha \to 1$. Mixing-in more online data thus increases the left hand side. Also note that the right hand side decreases due to $d^\mu(s)/d^\pi(s)$ if $\pi$ visits the state $s$ more often than $\mu$. Furthermore the larger the action value gap in the real Q-function $Q(s,a^*) - Q(s,b)$ the lower the right hand side. Finally the denominator will be negative if $\max_{b \notin A^*}[Q^\omega(s,b)] < Q^\omega(s,a^*)$ thus enabling correct learning even in the pure off-policy case with $\alpha = 0$.

Note that all of those conditions can be computed and checked if an accurate Q-function and state distribution is accessible. How to use imperfect Q-function estimates to adaptively choose such an $\alpha$ remain a question for future research.

**Proposition 4.** *Let $G_{\mathrm{IS}}^{\pi,\mu_z}$ be a set of importance sampling estimators as defined in equation 7. Note that they all have the same fix point $V^\pi$ and contract with at least $\gamma$. Then the contraction properties carry over to $V_{\mathrm{trusted}}^\pi$. In particular $|V_{\mathrm{trusted}}^\pi - V^\pi|_\infty \leq \gamma |V - V^\pi|_\infty$.*

*Proof.* Let us consider the set of importance sampling estimators as defined in 7 and note that they all contract to the same fixed point $V^\pi$ with at least $\left|\mathbf{E}_{\mu_z|z}\left[G_{\mathrm{IS}}^{\pi,\mu_z}(s)\right] - V^\pi(s)\right|_\infty \leq \gamma |V(s) - V^\pi(s)|_\infty$ for any state $s$.

By Minkowski's inequality the contraction properties of importance sampled Monte-Carlo bootstraps carry over to $V_{\mathrm{trusted}}^\pi$ which is a $p(z|\mu_z \in M_{\beta,\pi}(s_t))$ weighted average:

$$
\begin{aligned}
|V_{\mathrm{trusted}}^\pi(s) - V^\pi(s)|_\infty &= \left|\mathbf{E}_z\left[\mathbf{E}_{\mu_z|z}\left[G_{\mathrm{IS}}^{\pi,\mu_z}(s)\right]\Big|\mu_z \in M_{\beta,\pi}(s_t)\right] - V^\pi(s)\right|_\infty \\
&= \left|\mathbf{E}_z\left[\mathbf{E}_{\mu_z|z}\left[G_{\mathrm{IS}}^{\pi,\mu_z}(s)\right] - V^\pi(s)\Big|\mu_z \in M_{\beta,\pi}(s_t)\right]\right|_\infty \\
&\leq \mathbf{E}_z\left[\left|\mathbf{E}_{\mu_z|z}\left[G_{\mathrm{IS}}^{\pi,\mu_z}(s)\right] - V^\pi(s_t)\right|_\infty \Big|\mu_z \in M_{\beta,\pi}(s_t)\right] \qquad (14) \\
&< \mathbf{E}_z\left[\gamma |V(s) - V^\pi(s)|_\infty \Big|\mu_z \in M_{\beta,\pi}(s_t)\right] \\
&= \gamma |V(s) - V^\pi(s)|_\infty
\end{aligned}
$$

$\square$

**Proposition 5.** *Let $G_{\mathrm{Vtrace}}^{\pi,\mu_z}$ be a set of V-trace estimators (see equation 8) with corresponding fixed points $V^z$ (see equation 3) to which they contract at a speed of an algorithm and behaviour specific $\eta_z$. Then $V_{\mathrm{trusted}}^\pi$ moves towards $V^\beta = \mathbf{E}_{z|\mu_z \in M_{\beta,\pi}(s_t)}[V^z]$ shrinking the distance as follows $\left|V_{\mathrm{trusted}}^\pi - V^\beta\right|_\infty < \max_{\mu_z \in M_{\beta,\pi}(s_t)} |\eta_z(V - V^z)|_\infty \leq \eta_{\max} \max_{\mu_z \in M_{\beta,\pi}(s_t)} |(V - V^z)|_\infty$ with $\eta_{\max} = \max_{\mu_z \in M_{\beta,\pi}(s_t)} \eta_z$.*

*Proof.* Recall the contraction properties of a V-trace importance sampled Monte-Carlo bootstraps $G_{\mathrm{Vtrace}}^{\pi,\mu_z}$ being
$$
\left|\mathbf{E}_{\mu_z|z}\left[G_{\mathrm{Vtrace}}^{\pi,\mu_z}(s)\right] - V^z(s)\right|_\infty < \eta_z |V(s) - V^z(s)|_\infty
$$
for an algorithm and behaviour specific $\eta_z < 1$ for a $z$ dependent fixed point $V^z$ and for any bootstrap $V$. We then show that $V_{\mathrm{trusted}}^\pi$ moves towards the weighted average of fixed points $V^\beta = \mathbf{E}_{z|\mu_z \in M_{\beta,\pi}(s_t)}[V^z]$, since

$$
\left|V_{\mathrm{trusted}}^\pi(s) - V^\beta(s)\right|_\infty < \eta_{\max} \max_{\mu_z \in M_{\beta,\pi}(s_t)} |V(s) - V^z(s)|_\infty
$$

holds for any bootstrap function $V$ as we show below.

$$
\begin{aligned}
\left|V_{\mathrm{trusted}}^\pi(s) - V^\beta(s)\right|_\infty &= \left|\mathbf{E}_z\left[\mathbf{E}_{\mu_z|z}\left[G_{\mathrm{Vtrace}}^{\pi,\mu_z}(s)\right] - V^z(s)\Big|\mu_z \in M_{\beta,\pi}(s_t)\right]\right|_\infty \\
&\leq \mathbf{E}_z\left[\left|\mathbf{E}_{\mu_z|z}\left[G_{\mathrm{Vtrace}}^{\pi,\mu_z}(s)\right] - V^z(s)\right|_\infty \Big|\mu_z \in M_{\beta,\pi}(s_t)\right] \\
&< \mathbf{E}_z\left[|\eta_z(V(s) - V^z(s))|_\infty \Big|\mu_z \in M_{\beta,\pi}(s_t)\right] \qquad (15) \\
&\leq \max_{\mu_z \in M_{\beta,\pi}(s_t)} |\eta_z(V(s) - V^z(s))|_\infty \\
&\leq \eta_{\max} \max_{\mu_z \in M_{\beta,\pi}(s_t)} |V(s) - V^z(s)|_\infty
\end{aligned}
$$

$\square$

## APPENDIX D   DETAILED ATARI RESULTS

We display the Atari per-level performance of various agents at 50M and 200M environment steps in Table 2. The scores correspond to the agents presented in Figure 1. The LASER scores are computed by averaging the last 100 episode returns before 50M or respectively 200M environment frames have been experienced. Following the procedure defined by Mnih et al. (2015) we initialize the environment with a random number of no-op actions (up to 37 in our case). Again following Mnih et al. (2015) episodes are terminated after 30 minutes of gameplay. Note that Xu et al. (2018) have not published per-level scores. Rainbow scores are obtained from Hessel et al. (2017).

Table 2: Per level performance of various agents at 50M and 200M environment steps (see Figure 1).

| Game | LASER Shared (sweep at 50M) | LASER Shared (sweep at 200M) | LASER (no sweep at 200M) | Rainbow (no sweep at 200M) |
|---|---|---|---|---|
| alien | 18635.3 | 18277.3 | **35565.9** | 9491.7 |
| amidar | 1838.3 | 2695 | 1829.2 | **5131.2** |
| assault | 26027.1 | **40603.2** | 21560.4 | 14198.5 |
| asterix | **496735.0** | 240770 | 240090 | 428200 |
| asteroids | 232651 | **257420.1** | 213025 | 2712.8 |
| atlantis | **889934.0** | 866584 | 841200 | 826660 |
| bank_heist | 1333.1 | **1712.8** | 569.4 | 1358 |
| battle_zone | 66900 | **131880.0** | 64953.3 | 62010 |
| beam_rider | 80830.5 | **125795.2** | 90881.6 | 16850.2 |
| berzerk | 46651.6 | **64513.1** | 25579.5 | 2545.6 |
| bowling | 42.4 | 47.4 | **48.3** | 30 |
| boxing | 99.8 | 99.4 | **100.0** | 99.6 |
| breakout | **852.5** | 850.3 | 747.9 | 417.5 |
| centipede | 208008 | **409702.8** | 292792 | 8167.3 |
| chopper_command | 24814 | 727333 | **761699.0** | 16654 |
| crazy_climber | 160494 | 88818 | 167820 | **168788.5** |
| defender | 355447 | **369397.0** | 336953 | 55105 |
| demon_attack | 133557 | **138000.6** | 133530 | 111185 |
| double_dunk | 0.1 | **23.5** | 14 | -0.3 |
| enduro | 0 | 0 | 0 | **2125.9** |
| fishing_derby | 45.4 | **62.6** | 45.2 | 31.3 |
| freeway | **34.0** | **34.0** | 0 | **34.0** |
| frostbite | 5297.4 | 2230.8 | 5083.5 | **9590.5** |
| gopher | 86222.2 | 39721.2 | **114820.7** | 70354.6 |
| gravitar | 1360.5 | **2812.0** | 1106.2 | 1419.3 |
| hero | 30159.2 | 36510.6 | 31628.7 | **55887.4** |
| ice_hockey | 20.2 | **38.7** | 17.4 | 1.1 |
| jamesbond | 21663 | **60402.5** | 37999.8 | 19809 |
| kangaroo | 13932 | 14187 | 14308 | **14637.5** |
| krull | **9559.3** | 5743.6 | 9387.5 | 8741.5 |
| kung_fu_master | 65032 | 81792 | **607443.0** | 52181 |
| montezuma_revenge | 1 | 1 | 0.3 | **384.0** |
| ms_pacman | 6089.3 | **6890.7** | 6565.5 | 5380.4 |
| name_this_game | 25998.9 | **27910.7** | 26219.5 | 13136 |
| phoenix | 458355 | **628711.6** | 519304 | 108529 |
| pitfall | -0.2 | -0.2 | -0.6 | **0.0** |
| pong | **21.0** | **21.0** | **21.0** | 20.9 |
| private_eye | 100 | 100 | 96.3 | **4234.0** |
| qbert | 20283.8 | 24600.8 | 21449.6 | **33817.5** |
| riverraid | 24138.1 | 35491.5 | **40362.7** | 22920.8 |
| road_runner | 52942 | **63762.0** | 45289 | 62041 |
| robotank | 63.6 | **67.8** | 62.1 | 61.4 |
| seaquest | 1802.2 | **557213.3** | 2890.3 | 15898.9 |
| skiing | **-8904.8** | -8980.1 | -29968.4 | -12957.8 |
| solaris | 2222.4 | 3017.6 | 2273.5 | **3560.3** |
| space_invaders | 36071.4 | **53124.3** | 51037.4 | 18789 |
| star_gunner | 331327 | **602540.0** | 321528 | 127029 |
| surround | **9.8** | **9.8** | 8.4 | 9.7 |
| tennis | 0 | 0 | **12.2** | 0 |
| time_pilot | 77899 | **113603.0** | 105316 | 12926 |
| tutankham | 251.8 | 268.5 | **278.9** | 241 |
| up_n_down | 341988 | **368586.5** | 345727 | 125755 |
| venture | 0 | 0 | 0 | **5.5** |
| video_pinball | 513121 | 397451 | 511835 | **533936.5** |
| wizard_of_wor | 22280 | **45335.0** | 29059.3 | 17862.5 |
| yars_revenge | 145055 | 144370 | **166292.3** | 102557 |
| zaxxon | 50486 | **106862.0** | 41118 | 22209.5 |

