# OpenReview forum: "Off-Policy Actor-Critic with Shared Experience Replay"
_ICLR.cc/2020/Conference — Reject_

### Official Review · AnonReviewer2 · 2019-10-18
**Official Blind Review #2**

**Rating:** 6

**Review:**

The authors investigate off-policy actor-critic reinforcement learning where they want to make use of shared experience replay. Two approaches were suggested and compared. One was to mix replayed experience with on-policy data and the other was to create trust regions that only selects well-behaved behavioral distributions for state value estimation.
According to the authors the several experiments provide evidence that their algorithm achieves competitive or even state-of-the-art results in data efficiency. They underpin this with some theoretical analysis.


**Experience Assessment:**

I do not know much about this area.

**Review Assessment: Checking Correctness Of Derivations And Theory:**

I assessed the sensibility of the derivations and theory.

**Review Assessment: Checking Correctness Of Experiments:**

I assessed the sensibility of the experiments.

**Review Assessment: Thoroughness In Paper Reading:**

I read the paper at least twice and used my best judgement in assessing the paper.

---

> ### Author Response · Authors · 2019-11-12
> **Response to Official Blind Review #2**
>
>
> Thank you for your review.

---

### Official Review · AnonReviewer1 · 2019-10-23
**Official Blind Review #1**

**Rating:** 6

**Review:**

This paper aims to improve the efficiency of the actor-critic method. The authors first analyzed the cause of instability in the prior work, from the perspective of bias and variance. Two remedies were then presented: (i) mixing the experience replay with online learning; (ii) proposing a trust region scheme to select the behavior policies. The authors finally tested the proposed method on Atari games, and showed the better results, compared with the state-of-the-art methods.

In my opinion, the empirical results are impressive, and the authors also provided some insights for the motivation. Given the results on Atari games, this paper could be a great contribution on the actor critic methods. The propositions are presented to support relevant claims, while their significance seems a bit limited, and some further clarification is necessary. The authors also need to address a few confusing statements and missing details.

1. In Proposition 3, the authors claimed that mixing with on-policy data can reduce the bias. I checked the proof but did not find anything relevant. Also, what is the amount of bias reduced?
2. In Equation (1), could you provide a formal definition for "V"?
3. The authors claimed at the beginning of Section 4 that the trust region method was proposed to mitigate the bias and variance problem of V-trace. However, I did not see how this is reflected in Propositions 4 and 5. Is this statement only based on empirical results?
4. It was mentioned right below Equation (4) that "Observe how this inner expectation ... matches the on-policy return...". Could you provide a formal proof?
5. What are the hyperparameters for the 9 agents used in Figure 1? Also, how did you choose "b" in trust region?
6. A few notation issues / typo:
(1) it's -> its
(2) In Equation (5), should "z \in M_{\beta, \pi} (s_t)" be "\mu_z \in M_{\beta, \pi} (s_t)"?
(3) At the 2nd line of Page 7, should the content for the indicator function be "\beta (\pi, \mu, s_t) < b"?


**Experience Assessment:**

I have published in this field for several years.

**Review Assessment: Checking Correctness Of Derivations And Theory:**

I assessed the sensibility of the derivations and theory.

**Review Assessment: Checking Correctness Of Experiments:**

I carefully checked the experiments.

**Review Assessment: Thoroughness In Paper Reading:**

I read the paper thoroughly.

---

> ### Author Response · Authors · 2019-11-12
> **Response to Official Blind Review #1**
>
>
> Thanks for your review.
>
>
> Re 1: Proposition 2 emphasizes that the V-trace policy gradient with clipped importance sampling optimizes a wrong objective. In particular the policy gradient implicitly optimizes the target policy for a wrong Q function. We can compute how wrong this Q-function is in expectation. We provide a formula for a state action dependent distortion factor w(s, a) <= 1 in propositions 2 and 3. The factor distorts the Q functions in multiplicative way. When w(s, a)=1 there is no distortion at all.
>
> The question of how biased the resulting policy will be depends on whether the distortion changes the argmax of the Q function. Little distortions that don’t change the argmax will result in the same local fixpoint of the policy improvement. The policy will continue to select the optimal action and it will not be biased at this state.
> The policy will however be biased if the Q function is distorted too much. For example consider a w(s, a) that swaps the argmax for the 2nd largest value, the regret will then be the difference between the maximum and the 2nd largest value. Intuitively speaking the more distorted the Q, the larger will be the regret compared to the optimal policy.
>
> More precisely, the regret of learning a policy that follows a distorted Q is:
> Regret = Q(s, a_best) - Q(s, a_actual)  = max_b Q(s, b) - Q(s, a_actual)
> where
>  * a_best = argmax_a (Q, a) is the optimal action according to the real Q
>  * a_actual = argmax_a(Q(s, a) * w(s, a)), is the optimal action according to the distorted Q
>
>
> In proposition 3 we recall that mixing online data leads to a linear interpolation between real Q function and the implied Q function. In practice this moves each w(s, a) closer to 1.0. Given sufficient online data the argmax can be preserved.
>
> We have expanded section 2.3 in the paper and added further derivations to the appendix after Proposition 3.
>
> In particular consider the added equation 13 which provides interpretation on how to choose alpha such that the learnt policy will correctly choose the best action. One of the insights is that alpha may be small if there is a large action value gap between a_best and b.
>
> The provided conditions can be computed and checked if an accurate Q function and state distribution is accessible. Using imperfect Q function estimates to adaptively choose such an alpha remains a question for future research.
>
> In this paper we investigate different constant alpha values for their practical performance. We empirically show in Figure 2 that alpha as small as 1/8 results in stable learning performance.
>
>
> Re 2: We have clarified that V is the bootstrap value -- the previously estimated state value function.
>
> Re 3: Propositions 4 and 5 show that the trust-region value estimation operator is a sound operator that really obtains an improved estimate in expectation. We consider this as an essential condition and present it here for reference to show the correctness of our method.
>
> Re 4: We have added a derivation. In related matters we reference Degris (2012) around equation 1.
>
> Re 5: We present in Figure 2 that running a hyper-parameter sweep of 9 agents with shared experience replay is better than running a sweep with 9 separate agents.
>
> Page 8 states: “On Atari sweeps contain 9 agents with different learning rate and entropy cost combinations {3 · 10−4 , 6 · 10−4 , 1.2 · 10−3} × {5 · 10−3 , 1 · 10−2 , 2 · 10−2} (distributed by factors {1/2, 1, 2} around the parameters reported in Espeholt et al. (2018)).”
>
> The “b” parameter in the trust region was investigated by considering the values {1, 2, 4} on DMLab-30. The differences were minor such that we excluded them from the figure to improve readability.
>
> Re 6: Thank you very much for pointing this out. We have fixed this in the revision.

---

### Official Review · AnonReviewer3 · 2019-10-26
**Official Blind Review #3**

**Rating:** 6

**Review:**

This paper investigates off-policy actor critic (AC) learning with experience replay using V-trace. It shows that V-trace policy gradient is not guaranteed to converge to a local optimal solution. To mitigate the bias and variance problem of V-trace and importance sampling, a trust region approach is proposed to adaptively selects only suitable behavior distributions when estimating the state-value of a policy. To this end, a behavior relevance function (KL divergence) is introduced to classify behavior as relevant. The proposed learning method LASER demonstrates the state-of-the-art data efficiency in Atari among agents trained up until 200M frames. In all, this paper is well motivated and technically sound. The draft can be improved by making it more self-contained by providing a sketch of the proof rather than refer everything to the appendix. Also it might be helpful to provide a pseudocode of LASER to help readers better understand the technical details.

Other comments and questions:

1) When talking about the selection process, z is treated as a random variable. What is its distribution?
2) what does “very off-policy learning” mean?
3) In figure 3(left), why “LASER: shared + trust region” performs worse than “LASER: not shared”?
4) In proposition 3. Q^w should be explained in the main text.


**Experience Assessment:**

I have published in this field for several years.

**Review Assessment: Checking Correctness Of Derivations And Theory:**

I assessed the sensibility of the derivations and theory.

**Review Assessment: Checking Correctness Of Experiments:**

I assessed the sensibility of the experiments.

**Review Assessment: Thoroughness In Paper Reading:**

I read the paper at least twice and used my best judgement in assessing the paper.

---

> ### Author Response · Authors · 2019-11-12
> **Response to Official Blind Review #3**
>
> Thanks for your review.
>
> We have provided pseudocode in the appendix and made the paper more self-contained.
>
> Re 1: The random variable z indexes the set of policies for which we have saved sampled episodes in the experience replay: Consider uniform sampling of experiences from replay -- in that case, the random variable z indexes the previous policies mu_z=pi_t that saved data to the replay. Here pi_t is the target policy at training step t. In this case the distribution of z (equal to t) would be uniform as the experience replay is uniform.
>
> We also consider the case where experience is sampled uniformly from both agents id (in a parameter sweep) and training time (episode id).
>
> Re 2: We have reworded this term in the updated version. By “very off-policy” in the abstract we meant learning from replay generated by other agent instances. This stands in comparison to classic experience replay where agents learn from data that they have generated themselves and saved into a replay buffer.
>
> Re 3: We present an actor-critic algorithm that is robust to off-policy data. We have shown that off-policy data from other agents may have an adverse effect (left green curve in Figure 3) and deteriorate performance significantly. The proposed trust region is able to discard harmful data. This avoids negative interference. However the harmful data still occupies space in the replay and in the training batch (where the loss is zeroed out). This can be a slight disadvantage in certain circumstances if computational resources are limited. Note that the trust region agent trained with population based training (red curve in the right plot) obtains the best results of all considered experiments.
>
> Re 4: Thanks for the suggestion. We have added this.

---

### Public Comment · ~Michael_Dann1 · 2019-11-06
**Sample efficiency of shared replay agents**

Hi there,

One aspect of this paper that I was unclear on is how much experience the shared replay agents have access to. Does the sharing of experience between 9 agents mean that they are effectively exposed to 1.8B frames by the 200M frame mark? If so, is it entirely fair to compare against agents like Rainbow that strictly learn from 200M frames? Either way, your results are impressive, but since they’re likely to become the new benchmark for sample efficiency in Atari (at least in the 200M frame setting) I think it’s important to have clarity on this.

---

> ### Author Response · Authors · 2019-11-12
> **Re: Sample efficiency of shared replay agents**
>
> Thank you for the question. We have addressed it in the updated version of the paper. In Figure 1 we now also present a single agent that uses the same hyper-parameter schedule that was published by Espeholt et al. (2018). This agent obtains a score of 431% human normalized median across the 57 atari games, achieving a new state of the art in the single agent regime. The fastest prior agent to reach 400% is presented by Kapturowski et al. (2019) requiring more than 3,000M steps. This constitutes a 15x improvement in data-efficiency like-for-like.
>
> Comparing our single-agent and population (9 agents) results, we would like to point out that:
> 1) population training achieves higher performance (448% vs 431%) but indeed observes 9x more frames;
> 2) the single agent result used an optimised hyper-parameter schedule from Espeholt et al. (2018), while the population set up reflects the setting where a good hyper-parameter schedule is not known;
> 3) Like-for-like comparing population training with and without shared replay, we observe that sharing the replay leads to more efficient training (370% vs 233% at 50M steps per-agent).

---

### Decision · Program_Chairs · 2019-12-19

**Decision:**

Reject

**Comment:**

The paper presents an off-policy actor-critic scheme where i) a buffer storing the trajectories from several agents is used (off-policy replay) and mixed with the on-line data from the current agent; ii) a trust-region estimator is used to select trajectories that are sufficiently close to the current policy (e.g. in the sense of a KL divergence).

As noted by the reviews, the results are impressive.

Quite a few concerns still remain:
* After Fig. 1 (revised version), what matters is the shared replay, where the agent actually benefits from the experience of 9 other different agents; this implies that the population based training observes 9x more frames than the no-shared version, and the question whether the comparison is fair is raised;
* the trust-region estimator might reduce the data seen by the agent, leading it to overfit the past (Fig. 3, left);
* the influence of the $b$ hyper-parameter (the trust threshold) is not discussed. In standard trust region-based optimization methods, the trust region is gradually narrowed, suggesting that parameter $b$ here should evolve along time.

---

> ### Author Response · Authors · 2019-12-22
> **Re: Paper Decision**
>
> We presented two experiments on atari - in both LASER obtains state-of-the-art results:
>  * single agent vs. single agent on (LASER is 15x better than R2D2 at the 400% mark)
>  * population training vs. population training (LASER is 4x better than IMPALA at the 400% mark)
>
> Our state-of-the-art claims are *not* based on a single-agent training vs. population training experiment. The comparisons (see above) are indeed like-for-like and fair.